# Curcumin Nanoparticles as Promising Therapeutic Agents for Drug Targets

**DOI:** 10.3390/molecules26164998

**Published:** 2021-08-18

**Authors:** Hitesh Chopra, Protity Shuvra Dey, Debashrita Das, Tanima Bhattacharya, Muddaser Shah, Sidra Mubin, Samka Peregrine Maishu, Rokeya Akter, Md. Habibur Rahman, Chenmala Karthika, Waheed Murad, Naeem Qusty, Safaa Qusti, Eida M. Alshammari, Gaber El-Saber Batiha, Farag M. A. Altalbawy, Mona I. M. Albooq, Badrieah M. Alamri

**Affiliations:** 1Chitkara College of Pharmacy, Chitkara University, Punjab 140401, India; chopraontheride@gmail.com; 2Department of Food Science & Nutrition Management, J.D. Birla Institute, Kolkata 700020, India; protity.dey@gmail.com; 3School of Community Science & Technology, IIEST Shibpur, Howrah 711103, India; debashritadas95@gmail.com; 4Hubei Collaborative Innovation Center for Advanced Organic Chemical Materials, Ministry of Education Key Laboratory for the Synthesis and Application of Organic Functional Molecules, College of Chemistry and Chemical Engineering, Hubei University, Wuhan 430062, China; btanima1987@gmail.com; 5Department of Botany, Abdul Wali Khan University Mardan, Mardan 23200, Pakistan; waheedmurad@awkum.edu.pk; 6Department of Botany, Hazara University Mansehra, Mansehra 21310, Pakistan; shahhu123@gmail.com; 7Department of Biochemistry, University of Bamenda, Bamenda P.O. Box 39, Cameroon; perino7485@gmail.com; 8Department of Pharmacy, Jagannath University, Sadarghat, Dhaka 1100, Bangladesh; rokeyahabib94@gmail.com; 9Department of Global Medical Science, Yonsei University Wonju College of Medicine, Yonsei University, Wonju 26426, Korea; 10Department of Pharmaceutics, JSS Academy of Higher Education & Research, Ooty 643001, India; karthika1994haridas@gmail.com; 11Biochemistry Department, Faculty of Science, King Abdul Aziz University, Jeddah 80200, Saudi Arabia; nfqusty@uqu.edu.sa; 12Department of Chemistry, College of Sciences, University of Ha’il, Ha’il 2440, Saudi Arabia; squsti@kau.edu.sa; 13Department of Medical Laboratories, Faculty of Applied Medical Sciences, Umma Al-Qura University, Mecca P.O. Box 715, Saudi Arabia; eida.alshammari@uoh.edu.sa; 14Department of Pharmacology and Therapeutics, Faculty of Veterinary Medicine, Damanhour University, Damanhour 22511, Egypt; gaberbatiha@gmail.com; 15National institute of Laser Enhanced Sciences (NILES), Cairo University, Giza 12613, Egypt; f_altalbawy@yahoo.com; 16Department of Biology, University College of Duba, Tabuk University, Duba 71911, Saudi Arabia; malboog@ut.edu.sa; 17Department of Biology, Faculty of Science, Tabuk University, Tabuk 71491, Saudi Arabia; balamri@ut.edu.sa

**Keywords:** nanoformulation, curcumin, anticancer, *Curcuma longa*, turmeric

## Abstract

*Curcuma longa* is very well-known medicinal plant not only in the Asian hemisphere but also known across the globe for its therapeutic and medicinal benefits. The active moiety of *Curcuma longa* is curcumin and has gained importance in various treatments of various disorders such as antibacterial, antiprotozoal, cancer, obesity, diabetics and wound healing applications. Several techniques had been exploited as reported by researchers for increasing the therapeutic potential and its pharmacological activity. Here, the dictum is the new room for the development of physicochemical, as well as biological, studies for the efficacy in target specificity. Here, we discussed nanoformulation techniques, which lend support to upgrade the characters to the curcumin such as enhancing bioavailability, increasing solubility, modifying metabolisms, and target specificity, prolonged circulation, enhanced permeation. Our manuscript tried to seek the attention of the researcher by framing some solutions of some existing troubleshoots of this bioactive component for enhanced applications and making the formulations feasible at an industrial production scale. This manuscript focuses on recent inventions as well, which can further be implemented at the community level.

## 1. Introduction

Curcumin (C_21_H_20_O_6_) or [E,E-1,7bis[4-hydroxy-3-methoxy-phenyl]-1,6-hepadiene-3,5-ione] is the main phenolic pigment extracted from turmeric, the powdered rhizome of the herbs in the *Curcuma* genus [1]. It is a hydrophobic molecule, mostly insoluble in water (only 30 nM can be dissolved) and poorly soluble in hydrocarbon solvents [2]; however, curcumin is very soluble in polar solvents [3]. Several polar and non-polar organic solvents have been used in the extraction of curcumin including hexane, ethyl acetate, acetone, methanol, etc. Of the organic solvents employed, ethanol has been found to be the most preferred solvent for extracting curcumin. With its increasing use in dietary supplements, researchers are developing extraction methods by employing food grade solvents such as triacylglycerol to give good yields [4]. The diversity of available techniques/methods for the extraction of curcumin such as Soxhlet, microwave-assisted extraction, supercritical carbon dioxide extraction, ultra-sonic assisted extraction, enzyme-assisted extraction, zone-refining and dipping methods had been tried, and among these, the Soxhlet, ultrasonic, and microwave extractions are the most commonly employed methods [5].The microwave assisted extraction is based on heat targeting microscopic traces of moisture in a plant material. The moisture, when heated up inside the plant cell due to the microwave effect, evaporates and generates tremendous pressure on the cell wall causing the swelling of the plant cell [6]. The pressure pushes the cell wall from inside, stretching and ultimately rupturing it, which facilitates leaching out of the active constituents from the ruptured cells to the surrounding solvent, thus improving the yield of Phyto-constituents. This phenomenon can even be more intensified if the plant matrix is impregnated with solvents with higher heating efficiency under the microwave [7]. Curcumin is a thermo-labile entity which gets degraded after long exposure to microwave radiations [8,9]. Researchers demonstrated the selective and rapid extraction of curcuminoids from *Curcuma longa* (turmeric) into organic solvents using the microwave assisted extraction method and reported 60% extraction of curcuminoids with 75% purity within 1 min which relied on the selected solvent and exposure time to microwaves [10]. This extraction was optimized using acetone at 20% power level (PL). In another study, ref. [8] demonstrated the extraction of curcumin using formic acid and microwave assisted extraction from turmeric (*Curcuma longa* L.). They reported a 45.1 and 82.4% percentage purity of curcumin using formic acid and microwave assisted extractions along with the organic solvent, respectively [8]. The purity of the extracted samples was analyzed using the high performance liquid chromatography method. Although extraction with formic acid was fast, its purity efficiency was lower than purified curcumin from microwave assisted extraction. That was probably due to destruction of curcumin in formic acid. Therefore, acetone has been proposed as a preferable solvent for extraction instead of formic acid [8].

Several mechanisms/principles involved in ultrasonic assisted extraction have been identified. One such mechanism is the fragmentation attributed to the collisions between particles and ultrasonic waves that cause a reduction in the particle size, thereby facilitating mass transfer. Another is erosion which helps to improve the accessibility of the solvent by imploding the bubbles on the surface of the plant matrix. Sono-capillarity and sono-poration are able to improve the penetration of liquid through the channels produced by the bubble implosion and the alteration of the permeability of the cell membranes, respectively [10]. Srisath and researchers obtained an extraction yield of 72% of curcumin from *Curcuma amada*, a closely related species of *C. Longa* commonly known as “mango ginger” [11]. A green ultrasound process of preparation was developed which proved to be more efficient as compared to batch extraction and the operating temperature as compared to Soxhlet extraction [12]. The obtained yield was significantly higher with benefits of reduction in time. Ultrasound-assisted extraction can save energy and time, reduce extraction temperature and the amount of solvent, accelerate energy transfer, provide selective extraction, and increase productivity [13,14]. In Figure 1, the schematic representation of extraction techniques are well drafted by present reviewers. Conventional Soxhlet extraction is a traditional apparatus commonly used for the extraction of lipids and materials that are not water-soluble. Soxhlet can even store these substances, maintaining their properties. The drawbacks of the Soxhlet extraction process are enormous including being time consuming, being too laborious, and making use of the bulk amount of organic solvents [15]. Free from organic solvents, the supercritical carbon dioxide has been established in several countries for the extraction of curcumin, oleoresin, and curcuminoids from *Curcuma longa* and *Curcuma amada*. Carbon dioxide is inert, readily available, non-toxic, non-corrosive, non-flammable, recyclable, and it is stable at mild conditions under 37 °C temperature with a pressure of 72.8 bar which enables it to extract heat-labile compounds and also preserve their quality [16]. Normal operating conditions for this process are at pressures between 25 to 30 MPa and a temperature of 318 K. There are also a few reports on enzyme-assisted extraction, where pretreatment of turmeric with enzymes such as α-amylase and glucoamylase yielded a significant increase in curcumin yield [17].

## 2. Recent Trends in Curcumin Formulations Techniques for Effective Drug Delivery

Curcumin has been used in the treatment of several diseases including multiple myeloma, Alzheimer’s, psoriasis [18], and Anti-human immunodeficiency virus (Anti-HIV) cycle replication [19,20]. Some more medical applications of curcumin include skin and muscle anti-inflammatory, anti-bacterial, anti-carcinogenic, antifungal, antimicrobial agents, and recent prominent usage of curcumin is in anticancer compounds as a protective agent to minimize the risk of lung cancer, liver, duodenum, and kidneys [16]. In cancer therapy, curcumin has been shown to act on many signaling proteins, oncogenes, and transcription factors; it is also employed in the course of tumorigenesis, growth, and metastasis at different stages of carcinogenesis from the early effects that cause deoxy ribonucleic acid (DNA) mutations [21,22]. Curcumin arrests tumor growth by obstructing some key signal transduction pathways [23]. Despite its numerous beneficial effects, curcumin has some limitations such as low water solubility (hydrophobic), an unstable chemical structure, being rapidly metabolized but poorly absorbed in the body, as well as its utilization or bioavailability differing depending on the species and sex [24]. These obstacles of curcumin can be eliminated by synthesis of curcumin nanoparticles, liposomes, micelles, and phospholipid complexes which can be used for the purpose of longer circulation, permeability, and increased resistance to metabolic processes [25]. Chemicals modified by nanotechnology have been proven to be highly effective for drug delivery and targeting the required tissue [26].

Liposomes are lipid vesicles in the nano- to micro-size range which usually disperse in water. The vesicle membrane is composed of the lipid bilayer which has either one or more bilayers [27]. Liposomal properties, including size, lamellarity, bilayer rigidity, charge, and bilayer surface modifications, are influenced by lipid type, surface charge, and production method. Liposomes are useful in carrying drugs to the target sites, prolonging the drug duration of action, protecting drugs against degradation, protecting the patients against side effects or irritation, and solubilizing lipophilic compounds; therefore, poorly water soluble drugs have potential to be used as an injection [28]. Preparation methods for curcumin liposomes include the thin-film method, freeze-thawing method, injection method, reversed-phase evaporation method, etc. [25]. In the thin-film method, phospholipids, cholesterol, or other lipoids and fat-soluble drugs are mixed in an organic solvent and then evaporated on a rotary evaporator in a vacuum. When a uniform film appears, an aqueous buffer is added to hydrate the lipids to form liposomes [29]. The freeze-thawing method involves two procedures: first, freezing unilamellar vesicles that contain drugs. Second, liposomes are slowly melted to form a new stable kind of liposomes after a period of time [30]. Two injection methods exist based on the solvents used that are ethanol injection and ether injection methods. In the ether injection method, lipids and hydrophobic drugs are co-dissolved in an organic solvent as the oil phase. Then, the oil phase is rapidly injected into the water-soluble drugs with stirring. Liposomes are formed when the organic solvent is removed. However, the ether injection method is less frequently used in the preparation of curcumin liposomes because of the toxicity of ether as compared to ethanol which is less toxic [31]. In the reversed-phase evaporation method, phospholipids are dissolved in chloroform, ether, or another non-water miscible organic solvent. Then, the solution of the drug to be encapsulated is added and a w/o emulsion is prepared by means of short-time ultrasonic. After removal of the solvent by rotary evaporation, the residual inversed emulsion is diluted in the buffer solution. Gel filtration chromatography or ultracentrifugation can be used to separate unentrapped drugs from liposomes [32]. Chen et al. investigated the in vitro skin permeation and in vivo antineoplastic effect of curcumin by using liposomes as the transdermal drug-delivery system [33]. In the study [33], researchers reported a significant increase in drug permeation and the deposition of C-SPC-L (Curcumin-loaded Soybean Phospholipids Liposomes), C-EPC-L (Curcumin-loaded Egg Yolk Phospholipids Liposomes), and C-HSPC-L (Curcumin-loaded Hydrogenated Soybean Phospholipids Liposomes), respectively, compared to free curcumin solution. Moreover, in this study, C-SPC-L displayed the greater ability of all loaded liposomes to inhibit the growth of B16BL6 melanoma cells. Liposomal curcumin formulation has greater growth inhibitory and pro-apoptotic effects on cancer cells [31]. In Figure 2, the schematic diagram illustrates the mechanism of release of nanocurcumin which targets the lung carcinoma cells.

A nanogel is a nanoparticle (10 to 100 nm) composed of a hydrogel synthesized by either physical or chemical cross-linking of polymers under controlled conditions. The cross-linked structure of nanogels offers a strong base for drug storage and release. This technique prepares and releases active to cells for maintaining their activity, improving stability, and preventing drug immunogenicity [34]. Nanogels are developed as carriers for drug delivery and can be planned to absorb biologically active molecules spontaneously via the creation of salt bonds, hydrogen bonds, or hydrophobic interactions that can enhance oral and brain bioavailability of low-molecular-weight drugs and bio macromolecules [35]. These nanogels are designed as stimuli-responsive materials, which respond to changes in the pH, temperature, reductive environments, activity of enzymes, magnetic field, light, among others [36]. This response may cause changes in the conformation of the nanogels and can produce an “on-demand” triggered release of any loaded cargo. Nanogels’ characteristics can be finely regulated by changing their chemical composition [37]. A study revealed enhanced transdermal permeation of curcumin nanoemulgel against squamous cell carcinoma. The nanoemulgel showed significantly higher drug release and less toxicity [38]. A group of experts examined the in vitro cytotoxic activity of cell death of curcumin and nanocurcumin on the human breast adenocarcinoma cell line (MDA-MB231) [39]. Meristic acid-chitosan (MA-chitosan) nanogels were prepared by the technique of self-assembly. Curcumin was loaded into the nanogels. Khosropanah et al. 2016 found that curcumin-loaded nanogels were at least twice as potent as free curcumin, possibly due to enhanced uptake [39]. A literature on the stability and loading efficiency of the curcumin-loaded nanogels [40] showed that self-assembled nanogels obtained from hydrophobically modified dextrin are effective curcumin nano-carriers. It depicts that the formulation had a higher stability achieved in water than in the phosphate buffer saline, further evaluated by dynamic light scattering and fluorescence measurements. The report [40] also investigated the biological activity of the curcumin-loaded nanogels in HeLa cell cultures. The nanogels inhibited cancer cell growth as effectively as free curcumin.

## 3. Uses of Nanocurcumin

### 3.1. Wound Healing

Curcumin is basically a substance which is derived from the nature itself having innate properties and abilities of fighting against the microbial infections as well as healing the wounds. The overall process of healing of wounds is a dynamic as well as complex process comprising of various phases such as inflammation, proliferation, and maturation. However, the aqueous solubility of curcumin is very poor, and the curcumin’s profile of rapid degeneration deters the usage of curcumin. However, with the help of encapsulation of curcumin by nanoparticles, these kinds of hindrances can be successfully overcome, enabling topical delivery of the agent in an extended way. A schematic representation of the study of [41] is in Figure 3. It can be observed that the infected burn wounds which were treated with curcumin nanoparticles statistically showed a significant decrease in the counts of bacteria on the 3rd day and also the 7th day as compared to the control burn wounds which were not treated by the curcumin nanoparticle. It was also observed that the administration of nanocurcumin topically helped in faster healing of wounds than that of those which were not treated with nanocurcumin. Apart from the faster closure and healing of the wounds, the qualitative assessment in the study revealed the fact that the wounds treated by nanocurcumin showed formation of the granulation tissue in a better way, and also the re-epithelialization occurred earlier as compared to other wounds which were untreated. Nanocurcumin has an overall multifaceted impact on the healing of wounds and mostly on the proliferative phase it acts by increasing the deposition of collagens, production of fibronectin, as well as re-epithelialization. However, in this study, the toxicity of the nanocurcumin when applied topically in chronic wounds and their absorption and accumulation inside the internal organ has not been studied extensively. Wang et al. [34] pointed out that the curcumin solid lipid nanoparticles extensively exhibit better solubility properties than that of native curcumin. It helps in the downregulation of the pro-inflammatory mediators which are induced by polysaccharides (LPS) such as that of prostaglandin E_2_, nitric oxide (NO) etc. by the process of causing obstruction to the nuclear factor kappa-light-chain-enhancer of activated B cells (NF-κB) activation in the murine macrophage of RAW 264.7. The curcumin nanopolymers are used extensively in delivering the curcumin drugs effectively as it helps in improving the oral bioavailability along with better solubility of curcumin. Kartikeyan’s article [42] pointed out that the nanocurcumin polymers showed solid wound healing properties than that of free curcumin. On the other hand, ref. [43] showed that the nanocurcumin with solid dispersions helped in considerably improving the wound healing of the vaginal area because of improved bioavailability of the curcumin drugs which are very poorly soluble in water. It was encrypted in [44]’s article that nanocurcumin acts as a vital agent for fighting against inflammation by obstructing the activity of the enzymes and production of the cytokines and activating the factors of transcription. Moballegh et al. [45] showed that the nanocurcumin formulation of liposomes have an anti-inflammatory effect against 2-hydroxyethyl methacrylate present in the pulp stem cells of human teeth thereby improving the wound healing and dental care quality.

### 3.2. Hepatoprotective

According to [46], the curcumin nanoliposomes were prepared so that the size of the particle of curcumin could be reduced and also its solubility in water could be improved. As a result of this, the hydrodynamic layer surrounding the particles would be thinner with an increased rate of surface specific dissolution. It was mentioned that toxicity of the liver which includes necrosis and steatosis could be experienced as a result of any single exposure to the hepatotoxic agent CCl_4_. The study showed that the serum activities of alanine transaminase (ALT), Aspartate aminotransferase (AST), and alkaline phosphatase (ALP) were greatly reduced while the liver was treated with free curcumin as well as curcumin nanoliposomes. It was also noted that the treatment group receiving curcumin nanoliposomes treatment experienced lower serum activities of ALT, AST, and ALP as compared to that of the treatment group treated with free curcumin. This proved the fact that the curcumin nanoliposomes can act as the most beneficial agents for reversing any kind of injuries in the liver and exhibit the hepatoprotective effect successfully. It was also mentioned in the study that after being pre-treated with curcumin nanoliposomes followed by CCl_4_, the antioxidant enzymes such as catalase (CAT) superoxide dismutase (SOD) and glutathione peroxidase (GPx) were substantially reinstated to their basal level exhibiting better efficiency of curcumin nanoliposomes against that of peroxidation of lipids. On the other hand, in Alhusaini’s literature [47], nanocurcumin was used for the first time for managing the toxic effects of CuSO_4_ in liver tissues as it is believed that the smaller size of nanocurcumin could be especially helpful for readily interacting with the surface and inside biomolecules successfully. Maghsoumi et al. mainly carried out the study for evaluation of the protective as well as the regenerative effects of curcumin nanomicelles on the chronic liver injuries in mice which are induced by alcohol [48]. It was seen that irrespective of the little decrease in the liver enzyme levels in the group which was treated with nanocurcumin, the serum levels of AST, ALP, and ALT was reduced greatly in the post treatment groups of nanocurcumin. Thus, it was found in the study that nanocurcumin when administered at 100 mg/kg/day can be of utmost helpful in recovering the alcohol induced liver damages with bringing about a significant reduction in the lactate dehydrogenase (LDH) level. However, in this article, the clinical applications of the nanocurcumin were not conducted. In another study by [49], the hepatoprotective effect of nanocurcumin on the salinomycin induced liver toxicity in broiler chicken was studied. It was observed that the nanocurcumin addition in rations at a dosage of 200 mg/kg helped in substantially decreasing the enzyme level of AST. Liver enzymes such as AST are responsible for the metabolization of amino acids, and any increase in the levels of the enzyme indicates disease or damage of the liver. Sookoian et al. investigated the effect of nanocurcumin on protecting the hepatotoxicity caused in rats induced by carbon tetrachloride [50]. The study showed that nanocurcumin has a significant effect on protecting the liver in case of damage induced by carbon tetrachloride, and it also mentioned that the hepatoprotective effect of nanocurcumin is dependent on a particular dose of 2 mL/kg of body weight. In another article of [51], it was observed that in case of overweight or obese patients having non-alcoholic fatty liver disease, nanocurcumin helped in a significant increment of HDL and decreased the degree of fatty liver as well as liver transaminase which again highlights the hepatoprotective action of nanocurcumin successfully.

### 3.3. CVD

A group of diseases related to heart and disorders of blood vessels of the cardiovascular system which have a significant impact on the health and wellbeing of an individual is referred to as the cardiovascular diseases (CVD). In the article of [52], it has been mentioned that the curcumin nanomedicines helped in targeting of curcumin, improving cellular uptake and pharmacokinetics as well as efficacy of the nanocurcumin. The nanocurcumin also helped in improvement of circulation and enhancement of the permeation and retention of the therapeutic agent that has been loaded in the nanocurcumin. A study pointed out that there is generation of free radical oxygen species induced by doxorubicin causing the cardiac malondialdehyde level to rise up [53], thereby leading to extensive damage of the cardiac tissues by interacting with lipids, nucleic acids, and membrane proteins. Moreover, it was also seen that there was a significant decrease in the activity of cardiac superoxide dismutase and glutathione peroxidase in the group treated by curcumin conjugated nanoparticles. Thus, it was evident from this study that there are cardio protective actions showcased by the conjugated curcumin nanoparticles owing to their excellent antioxidant properties. Thus, the curcumin conjugated nanoparticles can be determined as a highly successful agent for limiting the cardiac organ injury by mopping up the free radical which is actually responsible for causing them. The patients receiving hemodialysis generally have greater susceptibility to suffer from CVD, and the morbidity in those patients is connected with higher C-reactive protein (CRP) levels and adhesion molecules such as vascular cell adhesion molecule 1(VCAM-1) and intracellular adhesion molecule 1(ICAM-1). The study of [54] investigated the impact of 120 mg of nanocurcumin on the levels of CRP and adhesion molecules for 16 weeks on the patients receiving hemodialysis. The end of study results pointed out the fact that the group receiving nanocurcumin treatment had a substantial decrease in the mean serum CRP level as compared to the group receiving placebo. The indication of the study clearly stated that there might be beneficial impact of nanocurcumin in decreasing the CRP levels and inflammation along with the adhesion molecules thereby having a cardioprotective effect. However, this study was conducted specifically for patients with hemodialysis and CVD related to hemodialysis, and the evidence provided in the study is still not sufficient enough for clinical trials. On the other hand, ref. [55] compared the effect of 500 mg curcumin capsule and 80 mg nanocurcumin capsule on 90 patients who have undergone coronary elective angioplasty. The results highlighted that as compared to curcumin, patients receiving nanocurcumin capsules showed better outcomes and changes in terms of level of total cholesterol, superoxide dismutase, malondialdehyde, triacylglycerol as well as tumor necrosis factor-alpha. Therefore, the nanocurcumin might show a better cardioprotective effect on the patients because of the higher bioavailability of nanocurcumin than curcumin. A study of curcumin loaded with resveratrol encapsulated together allowed for better solubility of nanocurcumin in the aqueous phase as compared to the drug alone [55]. It explored the cardioprotective impact of nanocurcumin in a cell model where cardiotoxicity was induced by doxorubicin. This showed a great reduction in the apoptosis as well as ROS in the case of rat embryonic cardiomyocytes which, when treated with doxorubicin hydrochloride, thereby pointed to the cardioprotective effect of the nanoformulation of curcumin. Another similar finding was seen in the study of [56] which established the fact that curcumin encapsulated in carboxymethyl chitosan nanoparticles helped in increased bioactivity and bioavailability. Moreover, the encapsulated nanocurcumin at a low dose of 5 mg/kg of body weight in a rat model showed a decrease in the cardiac hypertrophy.

### 3.4. Nervous System

In a living being, the nervous system is responsible for coordinating the behavior by the transmitting signal between various areas present inside the body. The central nervous system (CNS) and peripheral nervous system are two main parts of the nervous system in vertebrates which altogether are responsible for maintaining homeostasis and maintaining the reflexes of the spinal cord, helping in learning and memory and voluntary movement control. Wang et al. described the penetration of nanocurcumin to the blood brain barrier during cerebral damage after ischemic injury. The penetration of nanocurcumin causes inhibition of M1-microglial activation. Penetration of nanocurcumin causes accumulation at the ischemic penumbra [57]. In line with this study, ref. [58] found that the curcumin loaded poly lactic-co-glycolic-acid nanoparticles, when targeted to the neuroblastoma cell line for the treatment of Alzheimer’s Disease, showed inhibition of nuclear factor erythroid 2 related factor (Nrf2) which is a natural protein, thereby stopping oxidative damage of the cells. Moreover, it was also found that the expression of Apolipoprotein J or clusterin was decreased, thereby reducing the severity of the disease. Various types of curcumin nanocarriers, which include dendrimers, nanoparticle conjugates, micelles, and bio-based nanomaterials and their associated properties such as solubility, stability, etc., were studied in [59]. It was investigated that the nanocurcumin drug delivery system could actually lead to improved efficiency in the diseases related to CNS. Research gaps of the study are to be solved by further clinical studies so that nanocurcumin can be safely administered to the patients suffering from Alzheimer’s, Personality Disorder, and Amyotrophic lateral sclerosis. The exosomes which are primed with curcumin made by incubation of the brain endothelial cells in a mouse model for a time duration of 3 days helped in protection of the permeability of the endothelial cell layer by lowering the level of oxidative stress and mitigation of the action of junction proteins which have been impaired [60]. Researchers in [61] studied the neuromodulators effects of nanocurcumin and ω-3 fatty acids on the cycloxygenase 2 (COX-2) migraine network. COX-2 is an enzyme which is responsible for producing chemical messengers known as prostaglandins at a speedy rate which actually promotes inflammation. It was found in the results that both nanocurcumin and ω-3 fatty acids can help in reinforcement of their impact on COX-2 mRNA inhibition and also help in reduction in the levels present in the serum. Moreover, when used in a combined way, they were seen to reduce the severity, time duration, as well as frequency of headaches considerably in the case of migraine patients. The in vitro study of [62] determined that the novel curcumin loaded noisome nanoparticles were effective enough in the reduction of proliferation, viability, as well as migration of the glioblastoma stem like cells which have been collected from human glioblastoma multiform. The invasiveness of the glioblastoma stem like cell was also significantly inhibited most likely by inhibiting monocyte chemoattractant protein-1 along with inhibiting the growth of the tumor by upregulating the ROS when compared to that of free curcumin. It was evident from the study that for the purpose of delivery of curcumin to the glioblastoma multiform, nano-noisome could be an ideal one for the drug delivery, although further in vivo investigations are required for implementing it in clinical trials.

### 3.5. Lipid Profile

The lipid profile level in the blood consists of the total cholesterol, high and low density lipoprotein cholesterol (HDL and LDL), and triglycerides. Murthy et al. prepared nano-micelle containing curcumin for oral use, and in the clinical trial of the study, it was found that the nanocurcumin acts as a potent agent for the reduction in the different parameters of the lipid profile [63]. In addition to this, it was also found that the administration of nanocurcumin orally at a dose of 1gm/day could significantly help in the reduction in the concentration of triglyceride levels in the blood in the case of obese individuals. In the case of each subject of the group receiving nanocurcumin treatment, it was seen that after the treatment, the level of total cholesterol, triglyceride, LDL-C, and HDL-C was substantially reduced in the serum level than that of before treatment. Shamsi et al. studied the impact of various doses of nanocurcumin on the level of lipid profile present in the blood serum [64]. It was found in the study that the cholesterol, low density lipoprotein, very low-density lipoprotein, as well as triglyceride level in the serum of the diabetic rats were reduced to a great extent. Moreover, it was also established from the study that the mean level of high density lipoprotein in the serum was significantly increased in the group receiving nanocurcumin treatment. Both these studies proved that nanocurcumin treatment has a vital role to play in maintaining the lipid profile level in the serum. In line with these studies, ref. [65] studied the effect of nanocurcumin for short term supplementation in the case of girls who were overweight. But it was found that there were no such significant changes in the lipid profile of the overweight girls who were supplemented with nanocurcumin for a short time period. This study thus recommended using nanocurcumin supplementation for the long term and using various dosages for significant results on lipid profiles of overweight girls.

A magnificent analysis of the impact of nanocurcumin supplementation along with six weeks of high intensity interval training on the lipid profile of overweight girls was carried out [65]. It was determined that the group of overweight girls who received the nanocurcumin supplement as well as underwent training showed a significant amount of reduction in triglyceride levels and at the same time an increase in the high density lipoprotein level. However, the cholesterol and low density lipoprotein did not show any significant changes due to nanocurcumin supplementation. It was thereby established in the study that high intensity interval training together with nanocurcumin supplementation helped in showing an effect of reinforcement on each other in effectively maintaining the lipid profile level in overweight girls. In another literature by [66], it was found that the impact of nanocurcumin supplementation was much better in the case of animals having nicotine induced toxicity, and the ameliorative effect of the nanocurcumin normalized the level of cholesterol, triglyceride, LDL-C, VLDL-C and increased the level of HDL-C in the blood serum. All these literatures make it evident that nanocurcumin has important roles to play in properly maintaining the lipid profile levels in the body.

### 3.6. Antioxidant

The synthetic or natural substances that prevent or delay some form of damage to cells are known as antioxidants. Nanocurcumin can be categorized as a natural antioxidant having a potent bioactive agent. Some experts prepared a nanocurcumin suspension for testing its antioxidant activity using the 2,2-diphenyl-1-picrylhydrazyl (DDPH) method [67]. The result showed that the efficacy of such nanosuspensions was greater in antioxidant activity than pure curcumin or a simple mixture but not better than that of ascorbic acid. On the other hand, the study of [68] tried to establish the antioxidative as well as anti-diabetic results of nanocurcumin for the diabetic patient treatment. The results of the study showcased the fact that the activity of CAT, SOD, and GPX were significantly reduced in diabetic mice’s pancreas. The diabetic mice treated with nanocurcumin for about a period of 20 days showed a subsequent increment in the antioxidant with the help of reduction in the oxidative stress. The efficiency of oxidative stress reduction was also better in curcumin nanoparticles as seen in the study. In another study done by [69], the regulatory impact of nanocurcumin was evaluated against the injuries induced by tartrazine on the antioxidant status of rats. It was found that the liver and kidneys of tartrazine ingested rats which were treated with nanocurcumin showed a considerable increase in each of the antioxidant activities of the enzymes such as SOD, GPx, and CAT, and the levels of glutathione (GSH) and total antioxidant capacity (TAC) were also recovered with nanocurcumin administration to a great extent. A comparative study observed among curcumin with nanocurcumin the antioxidant activity of the mitochondria of liver using the aluminum phosphide (AIP) induced toxicity [70]. The results of the study highlighted the fact that the liver mitochondria experienced severe oxidative toxic effects after AIP exposure. However, huge improvements could be brought about in factors causing oxidative stress by treating with nanocurcumin. This in turn recommends that the administration of nanocurcumin could be beneficial in the case of the harmful impact of AIP induced toxicity of liver with the help of free radical scavenging and stabilization of the oxidative status of the liver. It was found out that the curcumin nanoencapsulation could be used as a highly effective way for combating the toxicity in the body induced by exposure to lead [71]. By administration of nanocurcumin along with lead, the reduced as well as oxidized levels of glutathione were restored along with bringing about a reduction in the ROS. The chelation property and better bioavailability of nanocurcumin was probably the actual reason for removing lead from the soft tissues and blood. Similar to this, the literature of [72] compared the defensive effects of nanocurcumin and curcumin against the injury caused in the lung induced by paraquat exposure. It was found in the study that the protective impact of nanocurcumin was much better than that of curcumin in the case of the prevention of injury of the lung induced by paraquat which was mostly by modulating the level of oxidative stress along with gene expression. Again, ref. [73] studied the antioxidant activity of curcumin nanocrystals against the circulatory toxic effects in Wister rats. The results of the study emphasized the fact that the efficacy of curcumin nanocrystals at a dose of 40 mg was better in the reduction in the lipid peroxidation level as well as increasing the antioxidant activities and detoxification of the enzymes such as superoxide dismutase, catalase, glutathione peroxidase, etc. These studies altogether highlight the efficacy of nanocurcumin as an antioxidant against various types of toxicity caused in the body.

### 3.7. Anti-Fibrinolytic Effect

The breaking down of a fibrin clot which is a coagulation product is obtained in the process of fibrinolysis. The main responsible enzyme in this regard is plasmin which cuts the fibrin mesh at different areas thereby producing circulating portions which are either cleared by liver, kidney, or other proteases. The literature of [74] compared the healing impact of curcumin and nanocurcumin which had been administered orally on the lacerated muscle. The literature mentioned that nanocurcumin is more effective, soluble, and bioavailable than that of curcumin, and it has vital role to play in regeneration of the muscle fiber along with reduction in the fibrosis. The nanocurcumin causes dissolution of fibrin and cellular migration due to upregulation of Urokinase plasminogen activator(uPA). Nanocurcumin is also very much useful in treating cases of submucosal fibrosis of the mouth owing to the fibrinolytic as well as anti-inflammatory feature of nanocurcumin. Therefore, taking into account these properties of nanocurcumin, it was estimated in this study that the supplementation of it orally can help in the reduction in the formation of scar tissue at the time of healing of the lacerated muscle. On the other hand, in the article of [75], it was found that Jun N-terminal kinase (JNK) along with mitogen activated protein kinase (p38 MAPK) help in the upregulation of the expression of the uPA gene with the help of curcumin which in turn is necessary in the promotion of cell migration along with fibrinolysis which can help in the healing of wounds. Considering the fact of better bioavailability, solubility of nanocurcumin, it could be estimated that nanocurcumin would be more effective in this regard for the fibrinolytic effect. In the literature of [76], it was reported that in the case of patients suffering from oral submucous fibrosis, curcumin as well as nanocurcumin showed anti-inflammatory properties with inhibition of the inflammation process. In addition to this, the fibrinolytic property of nanocurcumin was also proven by the lipid peroxidation inhibition along with the checking and reducing of the proliferation of cells. In line with these studies, it was shown in [77]’s study that exposure of bleomycin to the cells caused the activation of p53 phosphorylation thereby resulted in the increment of the Plasminogen Activator Inhibitor-I expression. Bleomycin plays a role in damaging the DNA in the Alveolar Epithelial Cells along with the increase in the regulation of cytokines, macrophages, neutrophils, and other inflammatory molecules. By treating with nanocurcumin, the p53 phosphorylation and PAI-I expression was shown to be reduced, thereby helping in the regulation of the fibro-proliferative injury phase which had been induced by bleomycin. Therefore, all these studies show the impact of nanocurcumin on fibrinolytic activity. Different types of nanocurcumin and their activities are shown in Table 1.

### 3.8. Anti Protozoal Activity

The protozoal diseases have become a huge burden (socially, economically and in terms of health and wellness) worldwide as infectious diseases. Some of the most common and life threatening protozoal diseases include—Malaria, African sleeping sickness, toxoplasmosis, Chagas’ disease, Giardiness, cutaneous and visceral leishmaniasis, and amoebic dysentery [85]. A study by [86] reported that in the tropical world, malaria is becoming a major concern in terms of public health, and statistically, the mortality rate was also observed to be higher. This increased effect of protozoan diseases is considered due to the lack of safe and cost effective vaccines and drugs for treatment [85]. Hence, the use of nanocurcumin in treatment of protozoan diseases is gradually gaining its importance as it has been reported to exert anti-protozoal activity. Firstly, curcumin is found to possess antimalarial activity when combined with chitosan, which increases its bioavailability and stability. Curcumin causes phosphorylation of cytokine level modulators showing its antimalarial effects. Similarly, it also controls the growth of trophozoites by exerting cytotoxic effects on *G. lamblia* trophozoites and is effective in treating human giardiasis, especially for anti-diarrheal purpose [87]. Curcumin has been found to be effective in alleviating infections caused by the protozoa *E. maxima* and *E. acervulina* in the upper and mid-section of the small intestine, as well as its alcoholic extract which has shown antiprotozoal activity against *E. histolytica*. One of the possible mechanisms that are considered for the antiprotozoal activity of curcumin is its effect on the gene transcription. Curcumin is found to have the ability to down-regulate NF-κB, which results in inhibition of Ikappabalpha kinase and thus reducing its phosphorylation. This in turn leads to induction of apoptosis, arrest in cell cycle, and suppressed growth of the cells which are infected by parasites. Moreover, it also inhibits the thioredoxin reductase, which further helps in control of the parasite proliferation. On the other hand, curcumin has also been found to inhibit certain molecules which help in the adhesion and establishment of species such as *Plasmodium* and *Toxoplasm* [88].

Another major concern related to protozoan diseases is the increase in drug resistance of the protozoa towards antiprotozoal medications. It has been found that these parasitic species undergo internal and genetic alteration and develop innovative mechanisms, which leads them to attain resistance from the drugs or therapies. Hence, it has successfully increased the need for a new and effective approach to reduce this resistance [85]. Once again, curcumin is found to be the solution to combat this problem. In the treatment of malaria, drug resistance of the Plasmodium strains is considered as a major threat. However, curcumin was found to be effective in sensitizing *P. chabaudi* and *P. falciparum*, which were resistant to drugs such as artemisinin and chloroquine, respectively. Glutathion transferase (GST), found in *P. falciparum*, is considered to be the reason behind its resistance to the drug chloroquine, and curcumin has been reported to inhibit the GST, thus becoming an useful agent in overcoming drug resistance in protozoan diseases [88].

### 3.9. Anti-Bacterial

When the proliferation of harmful strains of bacteria takes place, either on or inside the body, it leads to various infections and diseases. The use of antibiotics for the treatment of bacterial diseases and infections is commonly used since a long time. However, with time, the abuse of antibiotics has led to an increase in the resistance of the bacteria against the antibiotics, and this has become a major threat and serious concern for human health. Moreover, a misbalance is observed amongst the production of new antibiotics and the occurrence of bacterial strain with antibiotic resistance, which has further caused more challenges [89]. Drug resistance has become a global problem, with other limitations associated with it such as toxicity and additional health care expenditure [90]. In hospitals, Ryle’s tube is commonly used as a nutritional support and medication delivery system for many patients. The tube is passed through the nasogastric route. In a study by [91], it has been focused that these tubes are susceptible to bio-film forming bacteria such as *Staphylococcus* sp. per day. These bacteria start colonizing on the surface of the catheter, especially coagulase-negative *Staphylococci* (CoNS), and its surface attachment to abiotic or biotic surfaces occurs, which leads to further maturation and accumulation of the bio-film, and thus results in increased resistance to antibacterial agents. The two CoNS, which specifically exert such characteristics, are *S. haemolyticus* and *S. epidermidis* [91].

Curcumin is reported to have antibacterial properties and are used to treat infections since ancient times. The possible mechanisms of action concluded by scientists are—at first, it causes outflow of important metabolites of the bacterial cell membrane by forming an ionic channel or creating pores. Secondly, it disrupts the structure of the bacterial cell wall, leading to bacterial death [87]. Another research project reveals that in an in vitro experiment, curcumin was found to hinder the bacterial replication process by interacting with the prokaryotic filamenting temperature-sensitive mutant Z (FtsZ) protein and interfering with z-ring formation. Furthermore, the bacteriostatic action of curcumin by inhibiting the repair process of bacterial DNA is also considered as an antibacterial mechanism of curcumin [92]. A report showed evidence that curcumin is more effective in controlling Gram-positive bacteria as compared to Gram-negative bacteria [93]. Although the curcumin has potential antibacterial properties, its low solubility, low stability, and low bioavailability have resulted in decreased effectiveness of the antibacterial properties. Hence, when combined with nanotechnology, the effectiveness was found to be higher. On evaluation, it was found that due to the size reduction in the curcumin particles, higher toxicity and sensitization are observed in bacterial cells when compared to curcumin alone [94]. The use of nanocurcumin has also shown antibacterial activity against a broad range of bacteria such as *E. coli*, *S. aureus*, *P. aeruginosa*, and *B. subtilis*. Beside this, curcumin has exerted properties that improve the gut microbiota by retarding the growth of pathogenic gut bacteria and increasing the lactic acid bacterial count. As a result, it improved the integrity of the intestines [93]. Furthermore, a microbiological test delineated the efficiency of nanocurcumin against bacterial species such as *Escherichia coli, Staphylococcus aureus, Shigella dysenteriae*, and *Streptococcus pneumonia* was observed to be higher in comparison to amoxicillin, a commercial antibiotic [95]. On the other hand, an article by [91] stated that nanocurcumin is found to act as an excellent surface protecting agent and thereby inhibits the bio-film formation and CoNS colonization in nasogastric polyvinyl chloride (PVC) tubes. Thus, it is considered as useful antibacterial and anti-bio-film forming agents that can be applied to reduce and prevent medical device related infections.

### 3.10. Anti HIV

Human immunodeficiency virus (HIV) belongs to the Retroviridae family and is responsible for the development of acquired immunodeficiency syndrome (AIDS). AIDS has become a major public health concern since the time it was first reported in the year 1981 [96]. HIV reduces the immune system’s capacity to fight infections, and thus the person becomes susceptible to a wide range of infections that become life threatening at times and then gradually lead to AIDS development. Moreover, statistically it was found that an HIV infected person if left untreated reduces 9–11 years of the life expectancy [97]. HIV mainly targets the T helper cells (specifically CD4+ T cells), macrophages, and dendritic cells, which are considered as vital immune cells. The virus kills the cells by directly infecting it and also indirectly by the action of CD8+ cytotoxic lymphocytes, which cause cytotoxic activity on infected CD4+ T cells. Thus, lower numbers of CD4+ T cells are present which decreases the defense mechanism of the immune system. HIV can be transmitted by infected blood or its components (for example, using used needles) or by sexual contact [97].

Curcumin possesses anti-HIV activity, and it undeviatingly affects the protein molecules of the virus. The bioactive molecule binds directly in multiple target sites and as a result suppresses the activity of the enzymes. A review by [96] summarized that curcumin perfectly fits in the active site of the enzyme protease and when tested on a silico model, a complex of curcumin and boron showed that it had a therapeutic effect against HIV as it binds to different sites of the substrate-binding cavity of protease. When the cells gets infected, trans-activator of transcription (Tat) is secreted which further promotes the destruction of T-cells and stimulates the formation of HIV induced tumors. Curcumin is found to be effective against this as well, and it was observed that curcumin caused destruction of Tat by proteosomal degradation and suppressed the Tat acetylation which resulted in reduced HIV proliferation [96]. Similarly, ref. [87] outlined the anti-HIV activity of Cur-AgNP by downregulation of inflammatory mediators such as IL-1β, TNF-α, and IL-6, and also it caused hindrance in the replication of the HIV virus. Curcumin when combined with silver nanoparticles (AgNP) had been found to act as antiretroviral agents. HIV-1 is characterized by rapid replication, escapes the body immune defense, and also has genetic variability. Moreover, HIV-1 causes neurological complications due to its ability to increase the production of pro-inflammatory cytokines [98]. Furthermore, HIV-1 protease (PR) also helps in the production of new strains of viruses by the processing of poly-proteins. Hence, to reduce such complications, curcumin has been proven to be beneficial as it inhibits the activity of the HIV-1 proteases by binding to their active sites such as the CCR5 (C-Cchemokinereceptortype5). The CCR5 is a surface molecule of white blood cells and through this, the HIV-1 particles get entry in the body. Hence, the binding of curcumin molecules in CCR5 inhibits the entry of HIV-1, thus protecting the host cells from the HIV-1 virus [99]. In addition, due to the anti-inflammatory properties of curcumin, inhibition of pro-inflammatory cytokines is observed which further helps in the decrease in HIV-1 related complications [98].

### 3.11. Anti-Inflammatory

Failure to control inflammation results in chronic inflammation and serious tissue damage which becomes irreversible to cure and thus initiates various inflammatory diseases and in the long term results in serious metabolic diseases such as cardiovascular diseases, osteoporosis, and cancer [100]. Another major condition that is arising is the formation of sepsis. It has become a major health concern. When some infectious agents attack, it leads to severe inflammatory responses by activating Toll-like receptor (TLR) signaling pathways. The bacterial cells, which are responsible for infection, have LPS in their outer cell membrane. Excessive LPS binds to the Toll-like receptor 4 (TLR4) receptor and sends a signal for the release of pro-inflammatory cytokines and thus induces the inflammatory response [101]. Curcumin is found to hold strong anti-inflammatory properties. It is observed that curcumin blocks the production of cytokines and the activity of certain enzymes. Moreover, it obstructs the inflammatory response by the inhibition of certain signaling pathways of the nuclear factor, which leads to down-regulation of inducible nitric oxide synthase (iNOS) and COX-2 expressions [102]. Similarly, curcumin also negatively regulates certain adhesion molecules (such as VCAM, ICAM), IL-8, TNF, CRP, and IL-6. On the other hand, it increases the production and expression of IL-10 levels. The IL-10 is found to be a powerful anti-inflammatory agent that minimizes the autoimmune inflammation by controlling the immune activity. It also controls the production of pro-inflammatory cytokines such as IL-6, IL-12, and others [103].

The immune system plays a very important role in the inflammation process. Hence, to control that situation, modulation of the immune response should also be made. Curcumin has been proven to be a potent immunomodulatory agent and thus helps in the control of inflammatory process and as a result alleviates the tissue damage and inflammatory diseases. Curcumin controls the activation of B-lymphocytes, neutrophils, T lymphocytes, dendritic cells, and macrophages [103]. It also acts as an antioxidant, inhibiting production of ROS in the body and improving the immune functionality. Nanocurcumin was found to decrease the cytokine levels and as a result decreased the mortality and severity in patients with COVID-19. Hence, nanocurcumin has been suggested as a potential therapeutic agent in controlling inflammation or inflammatory diseases due to its anti-inflammatory and immune-modulatory properties [104]. Moreover, curcumin when loaded in solid-lipid nanoparticles have been found to be effective in treating LPS-induced sepsis by modulating the inflammatory pathways such as TLR4/2-NF-kB. Moreover, it is found to retard the release of pro-inflammatory cytokines in the same level as dexamethasone (a FDA approved drug for the treatment of sepsis) with lower side effects as dexamethasone, when used for a longer duration, has been found to cause adverse effects such as hypertension in the elderly and stunted growth in children [78]. Hence, by regulating the inflammatory pathways, nanocurmin has become a promising agent for the treatment of inflammatory diseases.

### 3.12. Anti-Tumor

The cause of the tumor formation is often stated to be due to an imbalance amongst the cell growth and cell death, where an increase in the former and decrease in the latter is observed. This, if left untreated, leads to cancer development [105]. Hence, to treat this disease, the medical field has developed different treatment processes, chemotherapy being one of the most common treatments. According to [106], the principal for the antitumor activity of the chemotherapeutics is by triggering the apoptosis pathway, causing impairment in the microtubules and causing DNA damage of the cancer cells. However, it is being observed that the cancer cells are getting resistance to the chemotherapeutic agents, which is often called termed as “chemoresistance”. The resistance of the cancer cells in induction of apoptosis is considered as one of the reasons of such phenomenon. Moreover, it is also observed that the chemo resistant cancer cells possess DNA protecting properties by stimulating DNA repairing pathways and thus preventing it from damage [106]. Curcumin is a phytochemical which is universally considered as a potential anticancer agent, and has been proven to provide protection to all types of cancer. In a disquisition done on mouse models, it was reported that curcumin inhibited the cell proliferation as well as stimulated the apoptosis cycle. It concluded on the note of using curcumin as a potent anticancer agent [107]. Similarly, an in vitro study found that curcumin, when administered in a dose-time dependent manner, was effective in inhibiting cancer cell growth in the lungs and increased their cell death. Along with anticancer properties, curcumin has also been proven to be effective against tumor formation by the process of autophagy. The downregulation of the PI3K/Akt/mTOR pathways results in autophagy. Moreover, the apoptosis induced by curcumin, along with the autophagy mechanism, synergistically causes toxicity for the cancer cells, leading to anti-tumor effects [105].

Curcumin is also found to suppress the cancer metastasis and decrease chemo resistance of cancer cells, along with reduction in drug side effects (such as cardio toxicity, hepatic toxicity, nephron toxicity and others). Cancer cells cause oxidative stress with a large amount of ROS formation. Thus, when curcumin is administered, due to its high antioxidant capacity, it reduces the oxidative stress in the body, and when co-administered with chemotherapeutic drugs, it exerts anti-tumor activity and also increases the drug sensitivity. Moreover, it hinders the colonization of cancer cells, which further provides better protection [106]. Curcumin has also been established to be effective in people suffering from colorectal cancer, and it was observed that it decreased the level of serum Tumor necrosis factor (TNF-α), as well as increased the apoptosis and also improved the overall health. However, due to its low bioavailability, the benefits are not utilized efficiently in the tumor site, and thus the importance of nanocurcumin is more focused due to its ability to enhance the benefits of curcumin as it increases the binding of the phytoconstituents in the target site and helps in treating the tumors [108]. The effect of nanocurcumin on breast cancer cell lines was investigated, and it was deduced that nanocurcumin was found to be an effective anti-tumor agent along with low toxicity [109]. Moreover, ref. [110] conducted a testing using a xenograft model in which it was found that nanocurcumin when used along with anticancer drugs showed a promising antiproliferative and anti-tumor effects in cancer cells which further helps in other cancer therapies. Figure 4 schematically represents the anti-tumor activity of nanocurcumin. Another advancement in nanodrug delivery is the use of nanocurmin combined with a magnetic field (NANOCUR-MF) of 8mT, which was reported to have a high anti-cancer, anti- tumor, and anti-microbial effect as in NANOCUR-MF; the solubility of nanocurmin is higher since the magnetic field helps in better membrane permeability [83].

### 3.13. Anti-Diabetic

Diabetes mellitus (DM) has become one of the most common chronic diseases globally and has become a huge financial burden. Hyperglycemia, alteration in carbohydrate, fat and protein metabolism, and glucosuria are some of the symptoms that are observed in such cases. Amongst all, type 2 diabetes is the most common in which the body develops resistance from insulin [111]. On the other hand, in type-1 diabetes, the pancreatic beta cells which are responsible for insulin production are either damaged or unable to produce enough insulin for the glucose uptake [112]. These dysfunctions result in high blood glucose levels which further leads to more complications in the body, one of them being Diabetic Sensorimotors Polyneuropathy (DSPN). In DSPN, the nerves of the legs and arms are usually affected which causes difficulty in motor activity and coordination and thus increases the chances of frailty and fall. Some people also suffer chronic pain which leads to anxiety, depression, and sleep disturbances. The main reason that is considered to cause diabetic neuropathy is the increased oxidative stress that the body undergoes due to the uncontrolled blood glucose level, increased ROS, and decreased endogenous antioxidant. Hyperglycemia also affects several other pathways such as the glucose auto oxidation, protein kinase C activation, polyol pathway, and hexosamine flux which activates the pro-inflammatory factors, ROS producing reactions, and nuclear factor kappa-light-chain-enhancer of activated B cells (NF-κB) leading to nerve damage [113]. Another major problem associated with diabetes mellitus is diabetic nephropathy. Diabetes mellitus is considered to be one of the leading factors of chronic kidney disease (CKD) and end-stage renal disease (ESRD). The high blood glucose increases the advanced glycated end-products (AGE) which activates protein kinase C (PKC) system and transcription factor NF-κB by interacting through the cellular receptors and results in inflammation and cellular damage. ROS also plays a major role in the development of diabetic nephropathy as it causes metabolic alteration of the molecules and thus brings changes in the renal hemodynamics [114]. Hence, the control of blood sugar to a normal range is very important to improve the overall health and quality of life.

Curcumin is considered as a medicine to control diabetes and the complications associated with it. It is found to improve insulin resistance by stimulating glycolysis and reducing gluconeogenesis metabolism in the liver. Moreover, the strong antioxidant profile alleviates hyperglycemia by reducing oxidative stress and ROS production [115]. It also stimulates the insulin production from beta cells and glucose transport in the blood. Certain biomolecules in curcumin are also found to modify the glucose transport protein structure and stimulate insulin receptors. When tested on rats, curcumin was found to exert regeneration and restoration of islets of Langerhans [116]. Streptozotocin (STZ) is a toxin, which is considered to cause oxidative stress and damage of pancreatic beta cells by apoptosis. A study concluded that nano curcumin is found to be effective against diabetes in rats induced by STZ. The nano curcumin decreased the inflammation and apoptosis of pancreatic beta cells and reduced the oxidative stress [82]. Mohiti et al. established that curcumin stimulates the glucose uptake of cells by enhancing GLUT4 translocation as well as increases the insulin sensitivity of the muscle tissues [117].

Moreover, curcumin has also shown beneficial effects on DSPN treatment. When experimented on mouse model, it was observed that nanocurcumin when administered for 4 weeks had resulted in a decrease in neuropathic pain due to its ability to inhibit tumor necrosis factor-α (TNF-α), NADPH oxidase-mediating oxidative stress, and (NO). It also had positive effects on the complications such as numbness, pain, vibration, and weakness [118]. Another review also reported improved fast blood sugar (FBS) and glycated hemoglobin (HbA1c) levels on induction of curcumin [51]. Hence, the antioxidant and anti-inflammatory properties of curcumin can be utilized to treat this chronic disorder.

### 3.14. Anti-Coagulant Activity

In many diseases, blood coagulation takes place inside arteries which may block the passage of blood supply or cause internal hemorrhage which can at times be life threatening. Hence, anticoagulants are usually administered in such cases to prevent the blood coagulation. Amongst many, cardiovascular diseases (CVD) are one of the most important disease in which the anticoagulants are commonly administered. In CVD, platelet aggregation and certain adhesion molecules are considered to be the main causative factors because these when not kept under regulation, it hampers the blood fluidity and gives rise to thrombosis. Moreover, inflammation also increases the chances of blood coagulation as it stimulates pro-coagulant molecules [119]. Similarly, blood coagulation is also observed in cancer patients and has become one of the important contributing factors of mortality and morbidity in them. In [120], researchers established that cancer is found to increase the chances of developing venous thromboembolism (VTE) by four to seven folds. On analysis of blood parameters of cancer patients, it was observed that most of them are in a hypercoagulable state and have different degrees of coagulation activation. Thus, it makes it clear that cancer patients are a vulnerable group to develop certain thrombotic conditions such as disseminated intravascular coagulation (DIC), venous or arterial thrombosis, and others. The pathophysiology of such a condition is the capacity of the tumor cells to stimulate the haemostatic system of the host, and this causes the cancer cells to express pro-coagulant proteins such as Cancer Procoagulant, Tissue Factor, and Factor VII, which further causes activation of blood clotting [120]. Another disease in which the problem of blood coagulation commonly associated is Chronic Kidney Disorders (CKD) and End Stage Renal Disorder (ESRD). Blood coagulating factors such as FVII, Fibrinogen, D-dimer, and FVIII are found to be in higher rates in CKD patients along with renal injury, which resulted in higher coagulation. Moreover, other secondary factors such as diabetes mellitus, CVD, hypertension, and others also cause a change in the hemostatic condition, as a result aggravating the thrombotic condition. The possible mechanism of increased coagulating factors are in CKD; renal damage takes place which leads to improper excretion of pro-coagulants. In addition, the secretion of pro-inflammatory cytokines results in higher inflammatory condition in the hosts, which activates the pro-coagulants [121]. Hence, the importance of anticoagulant is very crucial to reduce mortality and morbidity in patients.

In a pre-clinical research by [120], when curcumin was administered at a dose of 60 mg/kg to rats, it was found to reduce mortality by decreasing the intravascular coagulation. Curcumin reportedly decreased the platelet and leukocyte adhesion, as well as reduced the amount of plasma fibrinogen and TNF-α level in the blood. All the anticoagulant tests in the study concluded that curcumin coated films possess good anticoagulant activity, and it can be used for the treatment of arterial thrombosis [120]. Similarly, another study outlined the anticoagulant activity of curcumin as it was found to decrease the clotting time in (prothrombintime) PT and (activated partial thromboplatin time) APTT assays due to the presence of its hydrophobic groups. They inhibit the formation of thrombin. Curcumin also contains compound named ortho-methoxy group, which also helps in its anticoagulant activity. In addition to this, it has been found to inhibit the production of fibrin deposition in the kidneys and also lowered the white blood corpuscles (WBC) count which in turn results in lower platelet aggregation [122]. Hence, curcumin when combined with nanotechnology can be a promising agent as an anticoagulant and prevent thrombotic diseases.

## 4. Recent Patents for Nanocurcumin

There are various patents filed for the use or production of nanocurcumin; some of them are discussed below.

The major hurdle for use of curcumin is poor bioavailability. The curcumin has a high lipophilic nature and as a result it binds more to the body fat, leading to poor absorption from the intestine region and metabolism in liver later. After metabolism, it gets excreted through bile juices. The systemic bioavailability is low, and around 75% of what is ingested is excreted in the feces and urine. The US patent No. 20110190399A1 disclosed use of curcumin nanoparticles and chitosan nanoparticles for improving the bioavailability by 10-fold [123]. The bioavailability was evaluated by its capability to cure the *Plasmodium yoelii* infection in mice. The curcumin was loaded on the surface of chitosan nanoparticles. The olive oil was used as a delivery vehicle. However, the procedure does not cause any alteration in chemical structure and is certainly due to the availability of hydrophobic pockets on the surface of chitosan nanoparticles and helps curcumin in increasing the bioavailability and stability. Both of curcumin nanoparticles and curcumin bound to chitosan nanoparticles were able to cure 100% of the mice infected with the *Plasmodium yoelii* parasite in comparison to the control where all the animals died. The nanoparticles thus prepared can be suitable candidates for treatment of cancers, inflammatory reactions, Alzheimer diseases, cholesterol gall stones, diabetes, etc. The patent no. WO2010010431A1 discloses the method of preparation of self-nano-emulsifying curcuminoids mainly capable to deliver the high drug loading with improved bioavailability and has good physical and chemical stability [124]. The method uses a novel curcuminoid composition comprising curcumin or curcuminoids, a lipid carrier system with a hydrophilic-lipophilic balance (HLB) between 3 and 14, and a pH buffer, which forms a self nanoemulsion dilution with water, gastric fluid, or intestinal fluid of globule size of less than 200 nm. The liposomal form of nanocurcumin has an advantage that it has no QT (a part of ECG) prolongation activity, and is highly bioavailable but has rapid burst release. Whereas the polymeric nanocurcumin is high bioavailability, sustained or extended release and low in vivo clearance but prolongs the QT interval. The middle way, i.e., hybrid polymeric liposomes showed high bioavailability, sustained release and no QT prolongation, and low in vivo clearance. The US Patent No. US9138411B2, disclosed nanoparticles based composition based on the polymeric core comprising layer of one or more polymers and use of one or more active agents along with one layer of lipids on surface. The curcumin formulation thus formed was able to minimize QT interval prolonging for the treatment of cancer [125].

Only a few anti tubercular drugs are there, and Isoniazidis one of them. However, its use causes development of resistance in Mycobacterium, mainly caused by over expression of Inhibin alpha (Inh-A) or by mutations that causes lowering of the affinity to NADH. Moreover, prolonged administration causes the development of functional issue in the hepatic system and causes immune suppression. The world patent no. WO2014170820A2, disclosed pharmaceutical preparation containing the therapeutically effective amounts of nanocurcumin and isoniazid for the treatment of Tuberculosis [126]. The doses can be administered separately or simultaneously or in a fixed dose combination. The formulation modulates the T Helper cells’ response and prevents the death of T cells in patients. In an animal study, the effect of formulation was evaluated by measuring the spleenocytes in infected mice using ahaemocytometer. The mice administered with nanocurcumin showed a 70% increase, while the isoniazid caused a reduction in concentration by 25% while when both combined showed a 45% rise in the concentration of spleenocytes. In US patent US8747890B2, the pure curcumin (99.2%) was administered to the dogs at pre clinical toxicity studies, and it was found that dose dependent hemolysis was observed. The hemolysis prevented use of intravenous curcumin pure in treatment in neoplastic, parasitic, and tissue degenerative diseases. It was also found that RBC hemolysis is linked with Ca^++^ ions imbalance. The US Patent application no. US8747890B2 discloses the method to use the synthesized curcumin synergistically, i.e., diferuloylmethane and calcium channel blocker in subjects with neoplastic and neurodegenerative disorders [127]. The invention defines the effective amount of synthesized curcumin, and it is further enveloped by poly (lactic-co-glycolic acid) (PLGA). Copolymers to form liposomal formulation and one or more calcium blockers and helps in mitigation of curcumin induced RBC hemolysis. The US patent US8535693B2 disclosed the use of nanoemulsion topical formulation of curcumin, Tetrahydroxycurcumin, and curcuminoid either alone or in combination for treatment of inflammation, skin disorders, mucosal disorders, and related diseases thereof [128]. The curcumin has selective phosphorylase inhibitor, and it has been observed that suppressing the phosphorylase activity can lead to resolution of psoriasis. The patent no.US20180028447 described method of development of curcumin and piperine loaded double-layered biopolymer based nano delivery systems by using electrospray/coating method [129]. The core layer consists of zein protein, in which curcumin is encapsulated and the outer shell consists of chitosan in which piperine has been encapsulated. However, the clear mechanism highlighting the molecular mechanism of piperine for enhancement of curcumin has not been clarified, but it is shown that the residence period of curcumin has been increased by reducing the activity of Cytochrome P4503A4 (CYP3A4), which plays a role in the metabolism of curcumin. The nanoparticles thus formed can be targeted to distant tissues without compromising the stability and cellular ingestion efficiency and ultimately the bioavailability is reduced.

The patent no. US9555011B2 discloses the method of the formation of an active agent loaded activated PLGA nanoparticles for targeted cancer nano-therapeutics [130]. The composition included the method of making the activated polymeric nanoparticles for targeted drug delivery and consists of a biocompatible polymer and an amphiphilic stabilizing agent that is non covalently bonded with the spacer compound. Moreover, it has at least one electrophile that selectively reacts with nucleophile on the target agent. Similar results were also concluded via US Patent App. US20100290982A1 by preparing the solid in oil/water emulsion-diffusion-evaporation formulation for preparing curcumin-loaded PLGA nanoparticles [131]. The US9192644B2 describes the method of formation of bioavailable curcuminoid formulations for treating Alzheimer’s disease and other age-related disorders [132]. The curcumin can block the aggregation of Aβ and other amyloid-forming peptides as well as chelate metals that can cause lipid and protein DNA oxidative damage. Based on listed and other actions of curcumin, it has been suggested to use in the prevention or treatment of Alzheimer models. The formulations described here have been the described method of the formation of solid lipid nanoparticles or as encapsulated form with edible oils. The curcumin has been found to be less stable at a pH solution of above 7.0. However, the Tetrahydrocurcumin was more bioavailable nearly 7–8 times higher than pure curcumin and is stable at even basic pH. At pH 7 or above, the curcumin gets hydrolyzed into ferulic acid and vanillin. The combinations of derivatives of curcumin were evaluated, and Tetrahydrocurcumin was found to be more stable at physiologic pH. In addition, to prevent it from oxidation, the antioxidants such as ascorbic acid were used.

## 5. Recent Findings with Use of Nanocurcumin

Some of the latest research conducted in past years (2019–2021) have been cited in Table 2.

## 6. Research Gaps and Future Perspectives

Curcumin has received a broad-spectrum importance for therapeutic applications. An in-depth discussion of the present review report outlined the efficacy of curcumin as an effective drug candidate when nanotechnology is induced in modifying their physico-chemical properties. It is already discussed that this natural component has already managed a wide space due to its pharmacokinetic behavior. Every substance is not self-sufficient, thereby in this case also, there are drawbacks which are already brought to notice. Many questions and challenges are still not clarified and under consideration to declare it as an effective drug delivery agent. In this review, gradual changes of concept of traditional processing techniques and their new adaptation to nanoformulation are highlighted. Here, techniques such as freeze thawing, phase reversion are discussed, and present reviewers suggest some combination techniques to make it more feasible for industrial production. Some research gaps pointed so that extensive work can be carried out for wellbeing. We already know about signaling pathways of nanocurcumin in curing human diseases, but the human evaluation dosage should be of prime importance which is still at bay. Several conceptual preclinical studies are reported which should be cross verified by clinical trials in higher order animals. Toxicity related studies are scarce there by actual implementation is in a hazy state. Unfavorable toxicity of nanodrugs results in DNA damage, allergic response, and neuroinflammation, thereby extensive studies with accuracy are mandated and suggested. The tissue specificity reports for certain critical diseases such as tumor or cancers are also at the initial stage. So larger attention should be focused on the drug delivery system binding specific target. Efficacy of nanocomposites with curcumin is more effective compared to nanocurcumin or free curcumin. So, the researchers need to investigate for a conclusive report whether nanocurcumin or free curcumin can be used alone or in combination formulations as additional drugs. Researchers should extensively study on exact implications of nanocapsulated curcumin for further therapeutic or chemotherapeutic strategies.

## Figures and Tables

**Figure 1 molecules-26-04998-f001:**
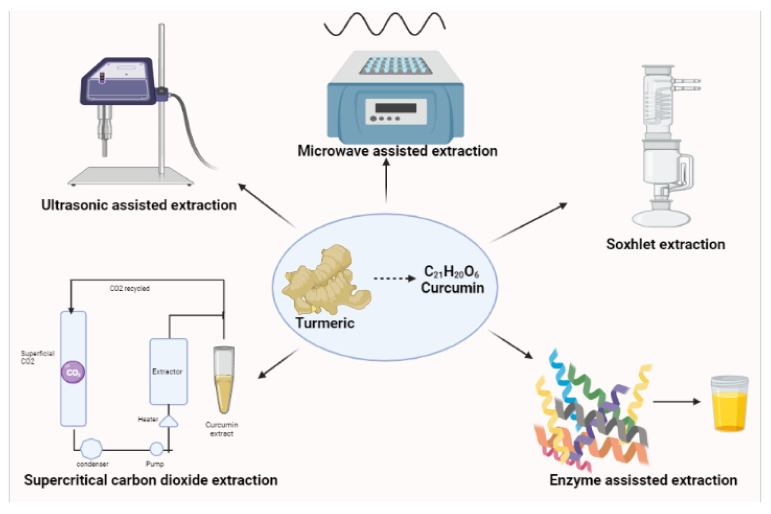
Schematic representation of curcumin extraction.

**Figure 2 molecules-26-04998-f002:**
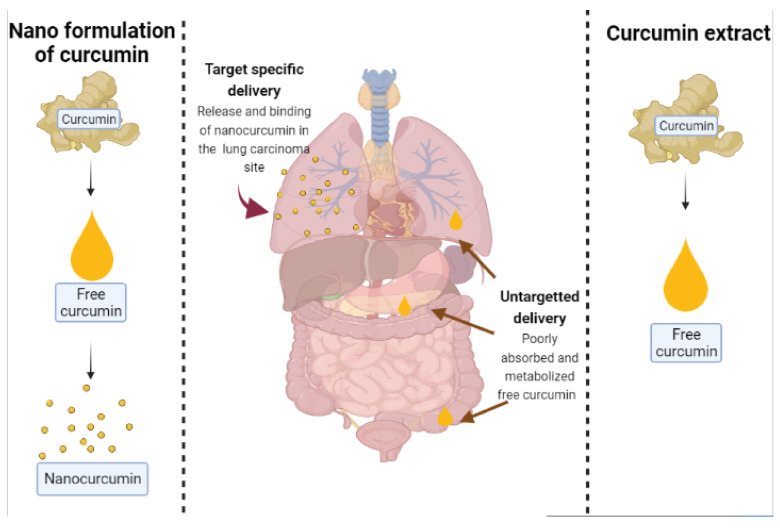
Difference in the delivery mechanism of free curcumin and nanocurcumin.

**Figure 3 molecules-26-04998-f003:**
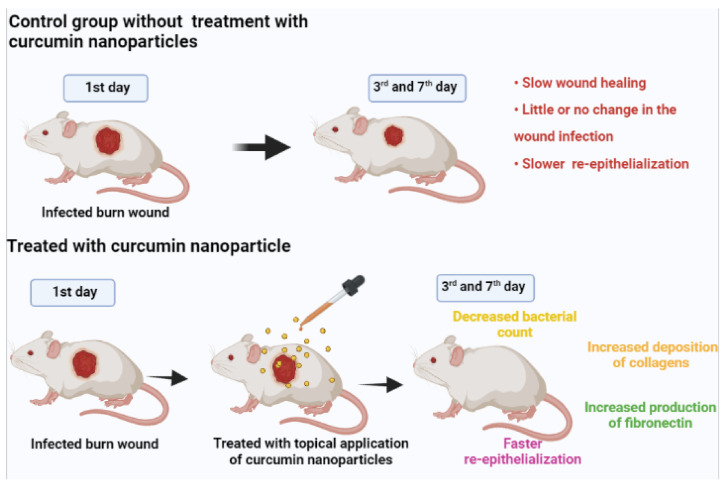
A schematic representation of wound healing activity of nanocurcumin study.

**Figure 4 molecules-26-04998-f004:**
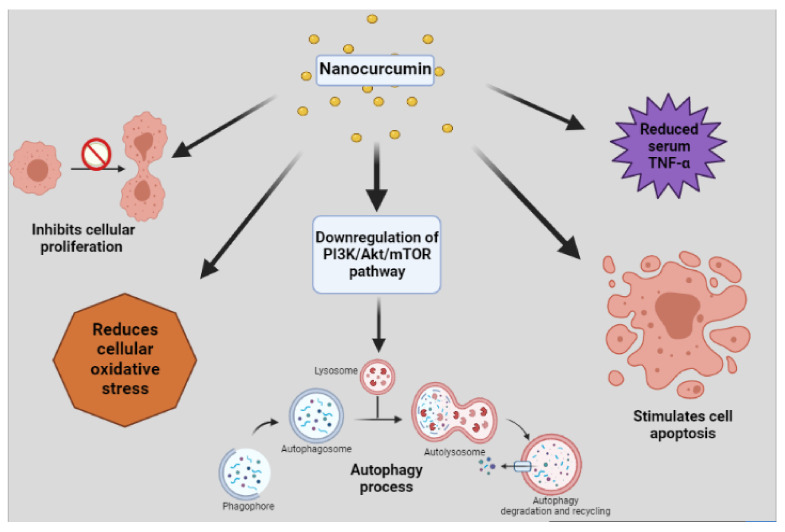
Schematic representation of anti-tumor activity of nanocurcumin.

**Table 1 molecules-26-04998-t001:** Different types of nanocurcumin and their activities.

	Size	Activity	Study Type	Reference
Curcumin loaded in Solid-liquid nanoparticles	60 nm	Prevents LPS induced sepsis in	Pre-clinical study	[78]
(Cur-SLNs)			Animal used: mice	
Zinc oxide–curcumin core–shell nanoparticles	~45 nm ZnO core and ~12 nm curcumin shell	Antibacterial activity	In vitro study	[79]
(ZnO–Cum)		(including the antibiotic resistant bacteria)	bacterial strains using the diffusion method	
Curcumin-TA-metal complex	200 nm 160 nm	Antibacterial activity	In vitro study	[80]
Cur@TA-Fe III Cur@TA-Cu II			Microbiological study on agar plates	
Nano-micelle containing curcumin (Sina Curcumin ^®^)	10 nm	Antidiabetic activity Decrease in insulin resistance Improvement in lipid profile	Pre-clinical study	[81]
			(study performed on diabetic rats)	
Nanocurcumin	300 nm	Antidiabetic Anti-inflammatory (STZ induced inflammation) Protects pancreatic beta cells	Pre-clinical study	[82]
			Animal used—rats	
NANOCUR-MF	34–359 nm	Anticancer Antimicrobial Antitumor	In vitro study	[83]
Nanocurcumin combined with magnetic field	8 MT magnetic field		(on mammalian cell line)	
Curcumin-reduced gold nanoparticles	26 nm	Anticancer Antitumor	In vitro study	[84]
(AuNP’s-Cur)			(on human cell line)	

**Table 2 molecules-26-04998-t002:** Literature review of various ground breaking research conducted using Nanocurcumin.

Disease Targeted	Outcome of Study	Reference
Coronavirus disease 2019	Nanocurcumin modulated increase in rate of inflammatory cytokines in COVID-2019 patients.	[133]
Coronavirus disease 2019	Symptoms of COVID-2019 resolved faster in group administered with Nanocurcumin and improved recovery rate.	[134]
Metabolic syndrome	Levels of Brain-derived neurotrophic factor, IL-10, serum concentrations of malondialdehyde decreased.	[135]
Oral lichen planus	Decrease in reticular-erosive-ulcerative (REU) score observed.	[136]
Knee Osteoarthritis	Reduced levels of Collagenase-2 and NO	[137]
Human lung cancer	Significantly inhibited the migration ability of A549 cells; promote intracellular ROS overproduction and induced apoptosis.	[138]
Migraine	Significant reduction in serum levels and expression of IL-17 mRNA	[139]
Migraine	PTX3 gene expression and serum levels were both significantly less	[140]
Human Glioblastoma	Enhancement in cytotoxicity against U87MG cell lines	[141]

## Data Availability

Not Applicable.

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
