# Peer review of "Curcumin Nanoparticles as Promising Therapeutic Agents for Drug Targets"

_molecules, 2021, doi:10.3390/molecules26164998_

Round 1

Reviewer 1 Report

It is not clear the reason for which in section 3, entitled “Nano curcumin formulation as anti-infective”, are comprised the following paragraphs:

3.1 Wound Healing

3.2. Hepatoprotective.

3.3. CVD.

3.4. Nervous system

3.5 Lipid profile

3.6 Antioxidant

3.7. Anti-fibrinolytic effect

3.12. Anti-tumor

3.13. Anti-diabetic

3.14. Anti-coagulant activity

More precisely, In paragraph 3.1, apart from the reduction in bacterial counts, described from row 219 to row 222, the majority of paragraph does not concern infected wounds.  In paragraph 3.2, instead of discussing some models of infective hepatitis (such as viral ones), which  would be expected in the section concerning “anti-infective” properties of curcumin nanoformulations, there is just a discussion concerning the use of curcumin nanosystems in chemical induced liver damage. Paragraphs from number 3.3 to 3.14 describe disease mechanisms not properly “infective”, such as CVDs or diabetes…

Rows 199-201:  the sentence “Curcumin was loaded into the nanogels. Khosropanah et al. 2016 found  that curcumin -loaded nanogels were at least twice as potent as free curcumin, possibly due to enhanced uptake [39]. A literature on the stability and loading efficiency of the curcumin-loaded nanogels [40]showed that self-assembled nanogels obtained from hydrophobically modified dextrin are effective curcumin nano-carriers” has been written with different fonts.

Rows 350-352: there are only some general considerations taken from ref.57, which is a review. Moreover, it is not described the role of nanoformulations in overcoming the blood-brain barrier to reach the terapeutic site.

Rows 429-431: the definition of antioxidants is not clear

Rows 483-484: the following sentence: ” The nanocurcumin is reported to have fibrinolytic properties by expressing the urokinase plasminogen activator (uPA)” is not clear and a little bit confusing.

Rows 499-501: the role of bleomycin in modelling the fibrotic deposition was not discussed

Row 520: curcumin is a very small molecule and it is not a kinase!

Rows 528-529: the following sentence: “ Curcumin is found to have the ability to down-regulate NF-κB, which results in inhabitation of Ikappabalpha kinase and thus reducing its phosphorylation” is not clear in describing the NF-kB pathway

Row 645: The sentence “Another major condition that arises from inflammation is the sepsis” is absolutely uncorrect on a scientific point of view. Indeed, sepsis is a medical emergency that describes the body’s systemic immunological response to an infectious process that can lead to end-stage organ dysfunction and death. Then, sepsis is a form of systemic inflammation but not all inflammatory disorders can evolve in sepsis

Rows 657-658: the sentence “The IL-10 is found to be a powerful anti-inflammatory agent that minimizes the autoimmune inflammation by controlling the immune activity”is out of context because it is focused on autoimmunity, which is not developed as concept all over the paragraph number 3.11

Row 833: please check the patent no. In the patent page the number is written  as US20110190399A1.

Row 835: The scientific name was not written italic.

Row 866: Ionized or Isoniazide???

Row 873: The preclinical toxicity studies with dog related to US Patent application no. US8747890B2? But can be easily confused with the previous patent No. WO2014170820A2. When shifting from one patent to another,  the information should be shared in such a way so that it can be easily understandable.

Rows 882-884 US patent 8,535,693. The ref. Is given as 128 (Di Martino RMC et. al). But the nanoemulsion topical formulation of curcumin is related to below author

Chaniyilparampu RN, Mungala M, Kapoor A, Gokaraju GR, Gokaraju RR, Kiran B, et al., inventors; Laila Pharmaceuticals Pvt. Ltd, India . assignee. Topical formulation(s) for the treatment of inflammation, skin and mucosal disorders and other diseases patent WO2010070675A2. 2010.

Di Martino RMC et. al just review the patent. The patent name was also written incorrectly. The correct name is US8535693B2.

Row 887 The patent number should be US20180028447/WO2016167732A1.

Rows 897-898 The patent no. US9555011B2 is related to below author-

Braden ARC, Vishwanatha JK. Formulation of active agent loaded activated PLGA nanoparticles for targeted cancer nanotherapeutics. US Patent 9555011B2, 2017.

But it is mentioned as ref no 130 which is related to Pandey P and Dureja H (2018) who are just reviewer. Please do the referencing correctly.

Row 902 its electrophile not electrophilic. If you one to use the word electrophilic then add ‘group’ with it.

Row 903 please recheck the patent no mentioned US Patent App. 12/766,068,??

Row 908 the sentence is confusing

 General the spacing and english sentence making should be more understandable.

Each reference does not contain the name of the journal, which is important for a review article to be mention.

Author Response

Cover Letter to the Editor’s and Reviewer’s Comments on Review Manuscript

Manuscript Title: Curcumin nanoparticles as promising therapeutic agents for drug targets

 

Journal: Molecules

Manuscript ID: 1311147

Dear Section Managing Editor,

Diana Fedorca

Thank you for your letter dated Jul 21, 2021. We are pleased to know that we can revise and resubmit our manuscript.

As per editor’s and reviewer’s comments, we made the following revision.

I am returning here with the above manuscript newly revised.

  • Necessary modifications has been made on the manuscript following reviewers comments.
  • According to reviewer’s comment everything has been corrected by authors and with the help of reviewers in the newly revised manuscript.

Appended to this letter is our point-by-point response to the comments raised by the editor’s. We would like to take this opportunity to express our sincere gratitude to the editor for the insightful comment. We would also like to thank you for allowing us to resubmit a revised copy of the manuscript.

Now we are anticipating this revised manuscript will be more suitable on your journal.

Cordially yours,

Md. Habibur Rahman and Muddaser Shah 

Reviewer 2 Report

The paper seems to be reliably prepared and contains an actual review of available data in the field.

I have only two remarks:

figures are the weakness of the paper, they are not informative, they do not provide useful information and do not look well. The Authors if they want to provide figures they should prepare better pictures. A nice example of the figures can be found here: https://www.ijpsonline.com/articles/multifaceted-role-of-a-marvel-golden-molecule-curcumin-a-review-3480.html

or here https://www.eurekaselect.com/141454/article

picture from this paper is presented below:

moreover the Authors should avoid the words like "magic" (line 922) or "elixir" (line 755). This should not be used in scientific papers.

Author Response

Cover Letter to the Editor’s and Reviewer’s Comments on Review Manuscript

Manuscript Title: Curcumin nanoparticles as promising therapeutic agents for drug targets

 

Journal: Molecules

Manuscript ID: 1311147

Dear Section Managing Editor,

Diana Fedorca

Thank you for your letter dated Jul 21, 2021. We are pleased to know that we can revise and resubmit our manuscript.

As per editor’s and reviewer’s comments, we made the following revision.

I am returning here with the above manuscript newly revised.

  • Necessary modifications has been made on the manuscript following reviewers comments.
  • According to reviewer’s comment everything has been corrected by authors and with the help of reviewers in the newly revised manuscript.

Appended to this letter is our point-by-point response to the comments raised by the editor’s. We would like to take this opportunity to express our sincere gratitude to the editor for the insightful comment. We would also like to thank you for allowing us to resubmit a revised copy of the manuscript.

Now we are anticipating this revised manuscript will be more suitable on your journal.

Cordially yours,

Md. Habibur Rahman

SL No

Reviewer -2 comments

Author’s reply

1

Figures are the weakness of the paper, they are not informative, they do not provide useful information and do not look well. The Authors if they want to provide figures they should prepare better pictures. A nice example of the figures can be found here: https://www.ijpsonline.com/articles/multifaceted-role-of-a-marvel-golden-molecule-curcumin-a-review-3480.html

or here https://www.eurekaselect.com/141454/article

picture from this paper is presented below:

New figures has been added as Fig-1-4.

Figure 1. Schematic representation of curcumin extraction.

Figure 2. Difference in the delivery mechanism of free curcumin and nanocurcumin

Figure 3. A schematic representation of wound healing activity of nano curcumin study

Figure 4: Schematic representation of anti-tumor activity of nanocurcumin

2

Moreover the Authors should avoid the words like "magic" (line 922) or "elixir" (line 755). This should not be used in scientific papers.

Both words has been removed from manuscript.

New Line no. 950-951:  It is already discussed that this natural component has already managed a wide space due to its pharmacokinetic behavior.

New Line no. 774-775:  Curcumin is considered as an medicine to control diabetes and the complications associated with it.

      Muddaser Shah and Md. Habib Ur Rahman 

Reviewer 3 Report

The paper Curcumin nanoparticles as promising therapeutic agents for drug targets, prepared by Hitesh Chopra et al., present novel and interesting review that deserve to be published after some minor improvements:

  1. the figures must be revised/re-drawn in order to be more attractive and professional.
  2. please add one more table that centralizes the recent findings in the field. Please summarize info from at least 15-20 papers published recently
  3. please try to cite other works related to curcumin, as follow and many others:

https://doi.org/10.33263/BRIAC103.490495

https://doi.org/10.33263/BRIAC95.225231

After proper revision, the paper deserves to be published.

Author Response

Cover Letter to the Editor’s and Reviewer’s Comments on Review Manuscript

Manuscript Title: Curcumin nanoparticles as promising therapeutic agents for drug targets

 

Journal: Molecules

Manuscript ID: 1311147

Dear Section Managing Editor,

Diana Fedorca

Thank you for your letter dated Jul 21, 2021. We are pleased to know that we can revise and resubmit our manuscript.

As per editor’s and reviewer’s comments, we made the following revision.

I am returning here with the above manuscript newly revised.

  • Necessary modifications has been made on the manuscript following reviewers comments.
  • According to reviewer’s comment everything has been corrected by authors and with the help of reviewers in the newly revised manuscript.

Appended to this letter is our point-by-point response to the comments raised by the editor’s. We would like to take this opportunity to express our sincere gratitude to the editor for the insightful comment. We would also like to thank you for allowing us to resubmit a revised copy of the manuscript.

Now we are anticipating this revised manuscript will be more suitable on your journal.

Cordially yours,

Md. Habibur Rahman

SL No

Reviewer -3 comments

Author’s reply

1

The figures must be revised/re-drawn in order to be more attractive and professional.

New figures has been added as Fig-1-4.

Figure 1. Schematic representation of curcumin extraction.

Figure 2. Difference in the delivery mechanism of free curcumin and nanocurcumin

Figure 3. A schematic representation of wound healing activity of nano curcumin study

Figure 4: Schematic representation of anti-tumor activity of nanocurcumin

2

Please add one more table that centralizes the recent findings in the field. Please summarize info from at least 15-20 papers published recently.

Table 2 has been added.

Disease targeted

Outcome of study

Reference

Coronavirus disease 2019

Nanocurcumin modulated increase in rate of inflammatory cytokines in Coronavirus-2019 patients.

[133]

Coronavirus disease 2019

Symptoms of Coronavirus-2019 resolved faster in group administered with Nanocurcumin and improved recovery rate.

[134]

Metabolic syndrome

Levels of Brain-derived neurotrophic factor, IL-10, serum concentrations of malondialdehyde decreased.

[135]

Oral lichen planus

Decrease in reticular-erosive-ulcerative (REU) score observed.

 [136]

Knee Osteoarthritis

Reduced levels of Collagenase-2 and NO

[137]

Human lung cancer

Significantly inhibited the migration ability of A549 cells; promote intracellular ROS overproduction and induced apoptosis.

 [138]

Migraine

Significant reduction in serum levels and expression of IL-17 mRNA

[139]

Migraine

PTX3 gene expression and serum levels were both significantly less

[140]

  Human Glioblastoma

Enhancement in cytotoxicity against U87MG cell lines

[141]

3

Please try to cite other works related to curcumin, as follow and many others:

https://doi.org/10.33263/BRIAC103.490495:

https://doi.org/10.33263/BRIAC95.225231:

Ref 141, has been added.

Arzani, H., Adabi, M., Mosafer, J., Dorkoosh, F., Khosravani, M., Maleki, H., Nekounam, H., Kamali, M. Preparation of curcumin-loaded PLGA nanoparticles and investigation of its cytotoxicity effects on human glioblastoma U87MG cells. Biointerface Res Appl Chem 2019, 9(5):4225-31.

Muddaser Shah and Md. Habib Ur Rahman

Round 2

Reviewer 1 Report

After the revision process, the review is suitable for the publication on  "Molecules".